# Inequalities in the care experiences of patients with cancer: analysis of data from the National Cancer Patient Experience Survey 2011–2012

Anna Bone,[1] Louise Mc Grath-Lone,[2] Sophie Day,[2] Helen Ward[2]

[1]School of Public Health, Imperial College London, UK
[2]Patient Experience Research Centre, School of Public Health, Imperial College London, UK

**Correspondence to**
Louise Mc Grath-Lone;
Louise.mc-grath-lone@
imperial.ac.uk

## ABSTRACT

**Objectives:** To explore inequalities in the care experiences of care by patients clinical or trust-level factors for patients with cancer.

**Design:** Secondary analysis of data from the National Cancer Patient Experience Survey 2011–2012.

**Setting and participants:** Adult patients with a primary diagnosis of cancer who attended an acute or specialist National Health Service (NHS) trust in England.

**Outcome measure:** OR of a patient rating their overall care positively, adjusting for other patient, clinical and trust-level factors.

**Methods:** Using cross-sectional data from 71 793 patients with cancer who completed the National Cancer Patient Experience Survey 2011–2012, we examined associations between patient, clinical and trust-level factors and a summary measure of patient experience, namely overall rating of care. Multivariate logistic regression was used to investigate variation by sociodemographic characteristics adjusting for other patient, clinical and trust-level factors.

**Results:** Female, non-white and younger patients were less likely to rate their overall care as excellent or very good. Patients with long-standing conditions, particularly those with learning disabilities or mental health conditions, also reported poorer overall care. This variation persisted when other patient, clinical and trust-level factors were controlled for, indicating that there are real differences in experiences among patients with cancer by sociodemographic characteristics.

**Conclusions:** There is evidence of inequalities in the experiences of patients with cancer in the UK by sociodemographic characteristics such as gender, age, ethnicity and disability. Quality cancer care services must strive to meet the needs of a diverse patient population equally; this study identifies patient groups for whom it appears cancer care services are in greatest need of improvement.

## INTRODUCTION

Patient experience is a key component of quality in healthcare and is one of the top priorities in the National Health Service

### Strengths and limitations of this study

- To the best of our knowledge, this is the first study to explore patients' overall rating of cancer care by sociodemographic characteristics including long-standing conditions or disabilities.
- A principal strength of this study is the large sample size (>71 000) and high response rate (68%).
- As this study involves secondary analysis of national survey data it is limited by the type of data available, for example, the influence of potentially important predictors of patient experience such as employment status, level of deprivation and health status could not be explored as these data were not gathered.
- A further limitation is that the binary categorisation of patients' responses condenses the patients' experiences (which were already limited to several multiple choice options) and may mask potentially significant variations.

(NHS).[1] However, studies in the UK indicate that there are systematic differences in patient experience by sociodemographic characteristics.[2–4] For example, studies on primary and hospital care have found that patients tend to report more positive experiences with increasing age,[4–8] females report less positive experiences than males[5 6] and non-white patients report less positive experiences than white patients, even after adjusting for other sociodemographic variables.[3 5 6 9] Less is known about variation in the experiences of patients with cancer. There are currently 1.8 million people living with cancer in the UK[10] and advances in cancer treatments mean that they are living longer and facing prolonged periods of contact with healthcare services because of complex treatment regimens.[11] In 2011, a department of health (DH) report set out the government's strategy to improve

outcomes by putting patients at the heart of cancer health services.[10] A key objective of this strategy was to reduce inequalities in care relating to both clinical outcomes and patient experience. Thus, exploration of the experiences of patients with cancer and the inequalities that may exist is critical in order to identify patient groups for whom cancer care services are in greatest need of improvement.

Detailed studies on the experiences of patients with cancer have often relied on small sample sizes.[12–14] However, a regular National Cancer Patient Experience Survey (NCPES) has been established which provides a wealth of information on care and treatment experiences. The 2011–2012 survey includes responses from over 71 000 patients with cancer from 160 trusts across the UK.[15] With a relatively high response rate and large sample size, the survey presents an opportunity to explore inequalities in the quality of care received by patients with cancer. Here, we aim to describe the variation in the experiences of patients with cancer by age, gender, ethnicity and presence of long-standing conditions or disabilities in order to explore whether there are systematic inequalities. We further examine the influence of clinical and trust-level factors on these variations.

## METHODS
### Source of data
We carried out a secondary analysis of 2011–2012 NCPES cross-sectional data collected by Quality Health (Chesterfield, UK) on behalf of the DH. All patients with a primary diagnosis of cancer who attended an NHS hospital as an inpatient or day case between 1 September 2011 and 30 November 2011 were sent the survey.[15] Non-responders were followed up with two postal reminders. The final response rate achieved was 68%. As no survey weights were available, the data could not be weighted to adjust for non-response. The dataset included demographic and clinical characteristics for 71 793 patients with cancer who attended 160 hospital trusts across England, as well as their responses to 70 multiple choice questions relating to various aspects of their experiences of care. Surveys such as NCPES are commonly used to measure patient experience over a range of domains; however, single summary measures of overall experience, such as the Family and Friends Test, have become increasingly important.[1] Our analysis focused on a summary measure of patient experience, namely patients' assessment of care as measured by Q70 in the survey, "Overall, how would you rate your care?" Responses from a five-point scale were transformed into a binary outcome, with 'excellent' and 'very good' categorised as 'positive' and 'good', 'fair' and 'poor' as 'not positive', in accordance with the DH Survey Guidance.[15]

### Patient, clinical and trust-level characteristics
The main sociodemographic characteristics of interest in this study (gender, age and ethnicity) were ascertained

by self-report[16] and grouped as in the national report.[15] As Chinese patients have reported less positive experiences than white patients elsewhere,[3 6 17] the 'Chinese' ethnic category was not combined with 'Other' in this analysis. For age and ethnicity the largest groups were chosen as the reference category. Responses to the question "Do you have any of the following long-standing conditions?" were used to identify patients with comorbidities. The clinical characteristics of tumour group and patient status (ie, day or inpatient) were taken from hospital administration records.

Patients with haematological cancer were assigned as the reference tumour group as the largest group (breast) did not have a representative age and gender distribution. Time since first treatment was ascertained by patients' survey responses. As trust-level factors have previously been associated with patient experience[6 18–20] several were included in this analysis. Hospital trusts were categorised by type (large acute, medium acute, small acute, specialist and teaching) and by foundation status. The Care Quality Commission's (CQCs) 2008/2009 Annual Health Check rating and the proportion of front-line staff satisfied with care at their trust (Q12d from the National NHS Staff Survey 2012) were also included as measures of trusts' overall quality. Quintiles of staff satisfaction were used as a categorical variable during regression analysis with the lowest quintile as reference category. The reference categories chosen for other trust-level factors were the largest groups.

### Data analysis
Variation in patents' overall rating of care by patient, clinical and trust-level factors was investigated using univariate logistic regression. Respondents with missing demographic, clinical or trust-level data or those who did not answer Q70, were then excluded (ie, complete-case analysis was undertaken) and multivariate logistic regression was used to describe associations between the individual demographic characteristics of interest and overall rating of care. Confounding by patient, clinical and trust-level factors was controlled for through their sequential addition to the model. Logistic regression was chosen as the small intraclass coefficient calculated for Q70 (<0.01) suggested the effect of clustering by trust among respondents was negligible; therefore, it was anticipated that a multilevel model and a multivariate logistic regression model would produce similar results.[21] However, as even small intraclass correlations can inflate type-1 errors, clustered robust SEs were utilised. All statistical analyses were conducted in Stata V.12.

## RESULTS
### Patient characteristics and rating of care
A total of 71 793 patients admitted to hospital trusts across England with a primary cancer diagnosis completed the survey. Table 1 shows the demographic and clinical characteristics of respondents and table 2 the

**Table 1** Characteristics of survey respondents and their unadjusted associations with a positive overall rating of care from univariate logistic regression

| | Patient characteristics | | | | | Clinical characteristics | | | |
|---|---|---|---|---|---|---|---|---|---|
| | n | Per cent | OR (95% CI) | p Value | | n | Per cent | OR (95% CI) | p Value |
| Gender | | | | | Tumour group | | | | |
| Male | 33 832 | 47.1 | 1 (ref) | | Brain/CNS | 746 | 1.0 | 0.52 (0.42 to 0.63) | *<0.001* |
| Female | 37 961 | 52.9 | 0.92 (0.88 to 0.96) | *<0.001* | Breast | 14 739 | 20.5 | 1.00 (0.92 to 1.09) | 0.98 |
| | | | | | Colorectal/lower GI | 9483 | 13.2 | 0.72 (0.66 to 0.78) | *<0.001* |
| Age group | | | | | Gynaecological | 4202 | 5.9 | 0.72 (0.64 to 0.80) | *<0.001* |
| 16–25 | 363 | 0.5 | 0.68 (0.51 to 0.92) | *0.01* | Haematological | 11 070 | 15.4 | 1 (ref) | |
| 26–35 | 969 | 1.4 | 0.62 (0.52 to 0.75) | *<0.001* | Head and neck | 2422 | 3.4 | 0.71 (0.62 to 0.81) | *<0.001* |
| 36–50 | 6802 | 9.5 | 0.70 (0.64 to 0.76) | *<0.001* | Lung | 5029 | 7.0 | 0.77 (0.69 to 0.85) | *<0.001* |
| 51–65 | 22 885 | 31.9 | 0.79 (0.74 to 0.83) | *<0.001* | Other | 1138 | 1.6 | 0.72 (0.59 to 0.86) | *<0.001* |
| 66–75 | 23 643 | 32.9 | 1 (ref) | | Prostate | 5831 | 8.1 | 0.70 (0.64 to 0.78) | *<0.001* |
| 76+ | 17 131 | 23.9 | 0.85 (0.79 to 0.90) | *<0.001* | Sarcoma | 2451 | 3.4 | 0.61 (0.53 to 0.69) | *<0.001* |
| | | | | | Skin | 1695 | 2.4 | 0.98 (0.82 to 1.16) | 0.80 |
| Ethnicity | | | | | Upper GI | 4540 | 6.3 | 0.61 (0.55 to 0.68) | *<0.001* |
| White | 63 652 | 88.7 | 1 (ref) | | Urological | 8447 | 11.8 | 0.64 (0.58 to 0.70) | *<0.001* |
| Mixed | 199 | 0.3 | 0.66 (0.45 to 0.97) | *0.04* | | | | | |
| Asian/Asian British | 1082 | 1.5 | 0.33 (0.29 to 0.38) | *<0.001* | Patient status | | | | |
| Black/Black British | 885 | 1.2 | 0.41 (0.35 to 0.49) | *<0.001* | Day case | 45 720 | 63.7 | 1 (ref) | |
| Chinese | 138 | 0.2 | 0.27 (0.19 to 0.39) | *<0.001* | Inpatient | 26 073 | 36.3 | 0.84 (0.80 to 0.88) | *<0.001* |
| Other | 510 | 0.7 | 0.58 (0.46 to 0.73) | *<0.001* | | | | | |
| | | | | | Time since first treatment | | | | |
| Long-standing conditions*† | | | | | <1 year | 44 997 | 62.3 | 1 (ref) | |
| None | 48 218 | 67.2 | | | 1–5 years | 17 486 | 24.4 | 0.83 (0.78 to 0.87) | *<0.001* |
| Deafness/hearing impairment | 7281 | 10.1 | 0.91 (0.85 to 0.98) | *0.01* | >5 years | 6212 | 8.7 | 0.88 (0.81 to 0.95) | *0.002* |
| Blindness/partially sighted | 1856 | 2.6 | 0.74 (0.65 to 0.84) | *<0.001* | | | | | |
| Physical condition | 9347 | 13.0 | 0.71 (0.67 to 0.76) | *<0.001* | | | | | |
| Learning disability | 354 | 0.5 | 0.50 (0.39 to 0.65) | *<0.001* | | | | | |
| Mental health condition | 1347 | 1.9 | 0.55 (0.48 to 0.64) | *<0.001* | | | | | |
| Long-standing illness‡ | 9241 | 12.9 | 0.77 (0.73 to 0.82) | *<0.001* | | | | | |

Total number of respondents=71 793. Ethnicity was unknown for 7.4% respondents, long-standing conditions status for 7.3% and time since first treatment for 4.3%. Significant associations at α=0.05 level highlighted in italics.
*6.7% of patients (n=4780) had >1 long-standing condition, therefore the column total exceeds 100%.
†Reference category for specific long-standing conditions is not having that condition.
‡Such as (but not limited to) HIV, diabetes, chronic heart disease or epilepsy.
CNS, central nervous system; GI, gastrointestinal.

**Table 2** Characteristics of trusts attended by survey respondents and their unadjusted associations with a positive overall rating of care from univariate logistic regression

| Trust-level characteristics | | | | |
|---|---|---|---|---|
| | n | Per cent | OR (95% CI) | p Value |
| Trust type | | | | |
| Small acute | 6240 | 8.7 | 1.23 (1.12 to 1.34) | *<0.001* |
| Medium acute | 16 677 | 23.2 | 1.07 (1.01 to 1.14) | *0.02* |
| Large acute | 25 850 | 36.0 | 1 (ref) | |
| Specialist | 3224 | 4.5 | 1.54 (1.36 to 1.76) | *<0.001* |
| Teaching | 19 802 | 27.6 | 1.03 (0.98 to 1.10) | 0.24 |
| Foundation status | | | | |
| No | 31 798 | 44.3 | 0.82 (0.78 to 0.85) | *<0.001* |
| Yes | 39 995 | 55.7 | 1 (ref) | |
| CQC trust quality rating (2008/9) | | | | |
| Weak | 3926 | 5.6 | 0.85 (0.77 to 0.94) | *0.001* |
| Fair | 18 482 | 26.2 | 0.97 (0.92 to 1.03) | 0.34 |
| Good | 28 425 | 40.3 | 1 ref | |
| Excellent | 19 748 | 28.0 | 1.17 (1.10 to 1.24) | *<0.001* |
| Frontline staff satisfied with care* | | | | |
| Mean | 63.5% | | Quintiles of front-line staff satisfied with care† | |
| Median | 62.7% | | 1 (lowest) | 1 (ref) | |
| Range | 35.3–94.0% | | 2 | 1.10 (1.03 to 1.19) | *0.01* |
| | | | 3 | 1.11 (1.03 to 1.19) | *0.004* |
| | | | 4 | 1.17 (1.09 to 1.26) | *<0.001* |
| | | | 5 (highest) | 1.35 (1.25 to 1.45) | *<0.001* |

Total number of respondents=71 793. CQC trust quality rating was unknown for three trusts (1.7% of respondents) and the proportion of front-line staff satisfied with care was unknown for 1 trust (1.7% of respondents). Significant associations at α=0.05 level highlighted in italics.
*Calculated from responses to Q12d from the National NHS Staff Survey 2012.
†Trusts were categorised into quintiles according to the proportion of staff responding positively to Q12d.
CQC, Care Quality Commission.

characteristics of the hospital trusts they attended. The majority of patients were white, female and >50 years old and there were substantial numbers with disabilities or other long-standing conditions. The most common tumour groups were breast and haematological cancers. Most respondents had started their treatment in the last year and were admitted to hospital as day case patients on their most recent visit. Most were treated in large acute trusts, trusts with foundation status and trusts rated 'Good' by CQC. The majority of patients (96.5%, n=69 276) provided a response to Q70 "Overall, how would you rate your care?" In total 87.8% rated their care as 'excellent' or 'very good' while the remaining 12.2% rated their care 'good', 'fair' or 'poor' (figure 1).

### Variation in rating of care by patient-level, clinical-level and trust-level factors

Unadjusted associations between patient-level, clinical-level and trust-level characteristics and Q70 from univariate logistic regression analysis are shown in tables 1 and 2. Among all respondents, statistically significant variation in overall rating of care by patient-level characteristics such as ethnicity, gender, age and long-standing conditions was observed. For example, women were less likely than men, and non-white patients were less likely than white patients, to rate their overall care very good or excellent. Chinese patients reported least favourably among non-white ethnic minorities. Younger

and older patients were less likely than 66–75-year-olds to rate their care very good or excellent, with the youngest patients (16–24-year-olds) least likely to report excellent or very good overall care. Patients with any long-standing condition were less positive about their overall care; those with a learning disability or mental health condition were the least satisfied.

Clinical-level and trust-level characteristics were also associated with overall rating of care. With the exception of patients with breast and skin cancer, all other patients

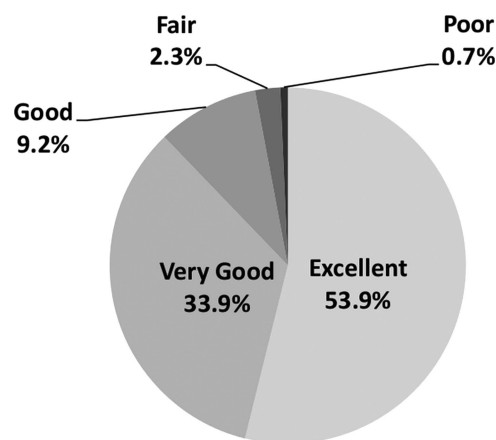

**Figure 1** Responses from NCPES 2011-12 to Q70 "Overall, how would you rate your care"?

were less likely than those with haematological cancers to rate their care as very good or excellent. Inpatients, patients who began their treatment more than 1 year ago and those who attended large acute trusts, trusts without foundation status or trusts with a 'weak' CQC rating were also less likely to rate their care as very good or excellent.

## Variation in patients' rating of care adjusting for clinical and trust-level factors

After excluding those with missing demographic, clinical or trust-level data or those who did not provide a rating of their overall care, 60 528 respondents from 150 trusts remained for complete-case analysis. The distribution of patient, clinical and trust-level characteristics in the 'complete-case' and 'all respondents' populations was similar (see online supplemetary table S1) and there was little difference in the univariate associations between the demographic characteristics and overall rating of care (with the exception of being deaf/having a hearing impairment which was not associated with a poorer rating of overall care during complete-case analysis, table 3). Model 1 in table 3 shows the effect of mutually adjusting for all patient-level factors. The observed variation in rating under univariate logistic regression was mostly unaffected; negative associations between rating overall care positively and being female, younger, non-white or having a long-standing condition persisted. The magnitude of the associations was generally stable though there was a slight increase in the effect of having a mental health condition or learning disability and being of mixed ethnicity was no longer significantly associated. The addition of clinical factors (tumour group, time since first treatment and in-patient or day-patient status) to the regression model (model 2) had little impact on variation by age or ethnicity, but the negative association between being female and care rating increased in magnitude. Including trust-level characteristics in the full multivariate model (model 3) had a minimal effect on the associations between patients' sociodemographic characteristics and rating of care. Even when adjusting for clinical, trust and other patient-level factors clear variation in patients' rating of care by sociodemographic characteristics such as gender, age, ethnicity and long-standing conditions was evident. Female, younger, non-white patients or patients with a long-standing condition remained less likely to rate their overall care as excellent or very good.

## DISCUSSION

Our analysis of the 2011–2012 NCPES demonstrates that there is marked variation in the experiences of patients with cancer by sociodemographic factors. Women, younger patients, ethnic minorities and patients with a long-standing condition or disability were less likely to rate their cancer care as 'excellent or "very good". This variation remained after adjusting for clinical factors,

such as tumour group and duration of treatment and trust-level factors. This suggests that the variation by sociodemographic factors is not a result of confounding but is attributable to real differences in experiences among these groups.

To the authors' knowledge, this was the first study to explore overall rating of care by patients with cancer by sociodemographic characteristics including long-standing conditions or disabilities. A principal strength of this study is the large sample size (>71 000) and the response rate of 68%, which was significantly higher than that achieved by comparable surveys.[22] [23] Also, as all patients with cancer treated by the NHS in England during the assigned 3 month study period were sent the survey, it is likely that findings can be generalised to the wider population of patients with cancer. The main limitations of this study relate to the type of data available. The data for the trust quality score was collected approximately 3 years prior to the NCPES survey period and so may not reflect the quality of the trust at the time of patient admission. The influence of other potentially important predictors of patient experience such as employment status,[20] level of deprivation[5] and health status[2] [8] [24] could not be explored as these data are not gathered through the NCPES. Furthermore, the binary categorisation of patients' responses, as per DH Survey Guidance, condenses the patients' experiences (which were already limited to several multiple choice options) and may mask potentially significant variations.

Interpretation of the findings from NCPES data requires consideration of the possible determinants of variation in patients' responses to a survey question. First, it is possible that variation reflects differing health, emotional or other support needs that are not met by cancer care services.[12] [14] [25] Second, differential expectations between patient groups, perhaps pertaining to sociocultural norms, may contribute to the observed patterns.[8] [26] Third, certain patient groups may have a tendency to respond less positively, based on shared norms regarding feedback and ideas as to its purpose.[26] Finally, variation may reflect real differences in the quality of care provided.[8] [26]

Studies of patient experience in the general patient population have demonstrated systematic differences in experience by gender, age and ethnicity and the results of our study further add to this knowledge by demonstrating that similar variation exists among patients with cancer. Adjusting for other sociodemographic factors, women were less likely to report positive experiences than men. This may be due to the increased emotional and support needs among female patients with cancer described elsewhere.[13] [14] Breast cancer was the most common tumour group for females (38.4%, n=14 591) and in comparison to other tumour groups patients with breast cancer were more likely to rate their care positively. Notably, when clinical factors such as tumour group were adjusted for, the magnitude of the negative association between gender and overall care rating increased. This may indicate that while patient

 

**Table 3**  Association between positive rating of overall care and demographic characteristics adjusting for patient-level, clinical-level and trust-level factors

| | Univariate* | | Multivariate<br>Model 1† | | Model 2‡ | | Model 3§ | |
|---|---|---|---|---|---|---|---|---|
| | OR (95% CI) | p Value | OR$_{adj}$ (95% CI) | p Value | OR$_{adj}$ (95% CI) | p Value | OR$_{adj}$ (95% CI) | p Value |
| Gender | | | | | | | | |
| Male | 1 (ref) | | 1 (ref) | | 1 (ref) | | 1 (ref) | |
| Female | 0.91 (0.87 to 0.96) | *<0.001* | 0.93 (0.88 to 0.98) | *0.02* | 0.72 (0.68 to 0.76) | *<0.001* | 0.72 (0.68 to 0.77) | *<0.001* |
| Age group | | | | | | | | |
| 16–25 | 0.65 (0.48 to 0.90) | *0.01* | 0.68 (0.48 to 0.98) | *0.04* | 0.61 (0.43 to 0.88) | *0.01* | 0.58 (0.41 to 0.82) | *0.002* |
| 26–35 | 0.67 (0.55 to 0.81) | *<0.001* | 0.71 (0.60 to 0.86) | *0.001* | 0.64 (0.53 to 0.77) | *<0.001* | 0.62 (0.51 to 0.75) | *<0.001* |
| 36–50 | 0.68 (0.63 to 0.75) | *<0.001* | 0.71 (0.65 to 0.78) | *<0.001* | 0.61 (0.56 to 0.67) | *<0.001* | 0.60 (0.54 to 0.65) | *<0.001* |
| 51–65 | 0.76 (0.72 to 0.81) | *<0.001* | 0.77 (0.73 to 0.82) | *<0.001* | 0.73 (0.68 to 0.77) | *<0.001* | 0.72 (0.68 to 0.77) | *<0.001* |
| 66–75 | 1 (ref) | | 1 (ref) | | 1 (ref) | | 1 (ref) | |
| 76+ | 0.85 (0.79 to 0.91) | *<0.001* | 0.86 (0.80 to 0.93) | *<0.001* | 0.90 (0.83 to 0.97) | *0.01* | 0.90 (0.84 to 0.98) | *0.01* |
| Ethnicity | | | | | | | | |
| White | 1 (ref) | | 1 (ref) | | 1 (ref) | | 1 (ref) | |
| Mixed | 0.65 (0.44 to 0.97) | *0.04* | 0.70 (0.45 to 1.10) | 0.12 | 0.68 (0.44 to 1.08) | 0.10 | 0.67 (0.43 to 1.04) | 0.08 |
| Asian | 0.35 (0.30 to 0.40) | *<0.001* | 0.37 (0.31 to 0.44) | *<0.001* | 0.35 (0.30 to 0.42) | *<0.001* | 0.35 (0.30 to 0.41) | *<0.001* |
| Black | 0.45 (0.37 to 0.53) | *<0.001* | 0.46 (0.38 to 0.56) | *<0.001* | 0.46 (0.37 to 0.56) | *<0.001* | 0.43 (0.36 to 0.53) | *<0.001* |
| Chinese | 0.26 (0.18 to 0.38) | *<0.001* | 0.28 (0.19 to 0.41) | *<0.001* | 0.29 (0.20 to 0.42) | *<0.001* | 0.27 (0.18 to 0.41) | *<0.001* |
| Other | 0.61 (0.47 to 0.77) | *<0.001* | 0.62 (0.48 to 0.79) | *<0.001* | 0.61 (0.47 to 0.79) | *<0.001* | 0.61 (0.47 to 0.77) | *<0.001* |
| Long-standing conditions¶ | | | | | | | | |
| Deafness/hearing impairment | 0.95 (0.87 to 1.02) | 0.16 | 0.93 (0.85 to 0.99) | 0.09 | 0.93 (0.85 to 1.01) | 0.09 | 0.93 (0.85 to 1.01) | 0.09 |
| Blindness/visual impairment | 0.78 (0.68 to 0.90) | *0.001* | 0.85 (0.74 to 0.97) | *0.01* | 0.85 (0.75 to 0.98) | *<0.001* | 0.86 (0.75 to 0.99) | *0.03* |
| Physical condition | 0.73 (0.68 to 0.78) | *<0.001* | 0.73 (0.69 to 0.77) | *<0.001* | 0.73 (0.69 to 0.77) | *<0.001* | 0.74 (0.70 to 0.78) | *<0.001* |
| Learning disability | 0.49 (0.38 to 0.65) | *<0.001* | 0.66 (0.49 to 0.88) | *0.01* | 0.68 (0.51 to 0.90) | *0.01* | 0.67 (0.50 to 0.90) | *0.01* |
| Mental health condition | 0.58 (0.50 to 0.67) | *<0.001* | 0.65 (0.57 to 0.74) | *<0.001* | 0.65 (0.57 to 0.74) | *<0.001* | 0.65 (0.57 to 0.74) | *<0.001* |
| Long-standing illness | 0.79 (0.74 to 0.85) | *<0.001* | 0.79 (0.74 to 0.85) | *<0.001* | 0.80 (0.75 to 0.86) | *<0.001* | 0.81 (0.75 to 0.87) | *<0.001* |

Significant associations at $\alpha$=0.05 level with clustered robust SEs highlighted in italics.
*Figures may differ from those presented in tables 1 and 2 as they are based on 60 528 respondents from 150 trusts with complete data (ie, complete-case analysis).
†Adjusted for patient factors (ie, ethnicity, gender, age group, specific long-standing conditions).
‡Adjusted for patient factors and clinical factors (ie, patient status, tumour group and time since first treatment).
§Adjusted for patient factors, clinical factors and trust-level factors (ie, trust type, foundation status, CQC trust quality rating and quintile of front-line staff satisfied with care).
¶Reference category for specific long-standing conditions is not having that condition.
Ref=reference category.

experience varies modestly overall by gender there are marked differences between men and women with less common cancers. This is an area which merits further exploration.

Younger patients were less likely to report positive experiences than older patients, which corroborates previous findings in relation to age and patient satisfaction.[4–8 13 27] It has been suggested that this observation may reflect a generational phenomenon, whereby older patients' responses are influenced by comparisons with their parents' generation who may not have had access to advanced technologies of modern treatment or the free care provided by the NHS, referred to as 'gratitude bias'.[28 29] Alternatively, younger patients may have higher expectations of quality of care due to a reduced frequency of hospital visits compared to older patients.[17] The poorer rating of care in the oldest age group (76+ years) fits with neither theory, hence further work to understand the cause of variation by age in the experiences of patients with cancer is required.

Ethnic minorities, especially Asian and specifically Chinese patients, reported less positive experiences than white patients. This trend is similar to findings from previous studies exploring variation in patient experiences of care generally[2 4 5] and specifically for cancer.[17 22] The extent to which these results are due to cultural differences in expectations of care or willingness to criticise is unclear and necessitates further research. Of significant concern is the possibility that these patients experience poorer quality of care owing to a lack of understanding of the care needs of these minority groups or to discrimination, unintended or otherwise.[8]

Patients with various long-standing conditions reported significantly less positive patient experiences than those without. The worst experiences were reported by patients with a learning disability or mental health condition. Given the small numbers of patients in these groups and the strength of the association it seems likely that there is marked variation in their experiences compared with other patients. Patients with long-standing illnesses such as diabetes and chronic heart disease were also less likely to rate their care as 'excellent' or 'very good.' Given that the number of patients with such illnesses is set to rise in the future with an ageing patient population it is important to explore how having comorbidities influences patients' experience of cancer care.

This study presents evidence of inequalities in experiences of cancer care by gender, age, ethnicity and disability. While it is possible that some of the variation observed between patient groups is a result of varying sociocultural expectations or tendencies to rate care positively, it is also possible that the quality of care truly differs between patient groups. Further investigation of the experiences of women, ethnic minorities, younger patients and those with a disability is needed so that cancer care services can be better tailored to meet the needs and expectations of these groups. Analysis of the NCPES qualitative free text questions and other patient experience data at a trust level would help to inform quality improvement initiatives. The findings of this study would appear to suggest that, if used as a comparative performance indicator (as is NCPES data), patient experience measures should be adjusted for age, gender and ethnicity. For example, an unadjusted measure of performance could unfairly disadvantage hospital trusts with higher than average proportion of ethnic minority patients . However, the impact of adjusting NCPES data for demographic characteristics on trust rankings has been shown to be minimal.[30] Adjusting for gender, age and ethnicity causes few trusts to move into or out of the top or bottom 20% of trusts nationally. While they may not account for much of the between-trust variation in the experiences of patients with cancer, the overall variation in patient experience by demographic factors is important in its own right and warrants further attention.

Responses to survey questions are a result of patients' perception and interpretation of events, which are shaped by expectations and clinical or emotional needs, in addition to the quality of services received. Meeting the care needs of all patients equally is a fundamental principle of the NHS and high-quality cancer services must strive to meet the needs of its diverse patient population. This study identifies patient groups for which cancer care services are in greatest need of improvement.

**Contributors** HW devised the study, advised on data analysis and participated in interpreting the data and reviewing the manuscript. SD participated in interpreting the data and reviewing the manuscript. LM carried out supplementary data analysis, participated in interpreting the data and coprepared the manuscript. AB carried out the statistical analysis, interpreted the data and coprepared the manuscript.

**Funding** HW, SD and LM are supported by the NIHR Imperial College Healthcare NHS Trust Biomedical Research Centre and Imperial College Healthcare Charity.

**Competing interests** None.

**Provenance and peer review** Not commissioned; externally peer reviewed.

**Data sharing statement** The National Cancer Patient Experience Survey 2012–2013 dataset used is available for download from http://ukdataservice. ac.uk.

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
