## [Reviewer comments · BMJ Open]

Some articles will have been accepted based in part or entirely on reviews undertaken for other BMJ Group journals. These will be reproduced where possible.

ARTICLE DETAILS

TITLE (PROVISIONAL)	Inequalities in cancer patients' experiences of care: analysis of data from the National Cancer Patient Experience Survey 2011-12
AUTHORS	Bone, Anna; Mc Grath-Lone, Louise; Day, Sophie; Ward, Helen

VERSION 1 - REVIEW

REVIEWER	Yana Vinogradova Nottingham University The United Kingdom
REVIEW RETURNED	13-Dec-2013

GENERAL COMMENTS	My main concern is missing values. About 1/6 of the questionnaires had missing values and were therefore removed from the multivariate analysis. At this level, the proportion of complete data should be reported in the text, not just in table footnotes. Some link should also be established between the univariate and multivariate analyses, because, as it stands, these analyses are effectively performed on two different samples. The number of missing values for ethnicity, long-standing conditions and time since first treatment are all not negligible at 7%, and it is not clear how removal of these observations might affect distributions for other variables. I would expect, therefore, some sensitivity multivariate analysis – either analysing missing values as separate categories or applying multiple imputations (whichever is more appropriate).
---

REVIEWER	jane maher Mount Vernon Cancer centre Northwood Middlesex HA6 2RN CMO Macmillan Cancer Support Macmillan Cancer Support has contributed funding to national patient experience surveys through the national cancer survivorship initiative.
REVIEW RETURNED	16-Dec-2013

GENERAL COMMENTS	The national patient experience survey suffers from all the limitations of national surveys of this sort but the high response rate has remained solid through repeated surveys. The study only looks at acute episodes of care and has a lower than prevalent response rate from classic "hard to reach" groups. It is not a new finding that there are inequalities in reported patient experience across gender, age and ethnicity but the fact that a national survey of this type is able to pick this up and demonstrate variation is of interest and I think publishable, I think worthy of publication.
--

VERSION 1 – AUTHOR RESPONSE

REVIEWER 1

Comment 1: My main concern is missing values. About 1/6 of the questionnaires had missing values and were therefore removed from the multivariate analysis. At this level, the proportion of complete data should be reported in the text, not just in table footnotes.

Response: Accept this point. The proportion of complete data is now reported in the text.

Changes made: Results added to [p11 , line 37-42]:

After excluding those with missing demographic, clinical or trust-level data, or who did not provide a rating of their overall care, 60,528 respondents from 150 trusts remained for complete-case analysis.

Comment 2: Some link should also be established between the univariate and multivariate analyses, because, as it stands, these analyses are effectively performed on two different samples.

Response: Accept this point. We have recalculated the univariate associations in Table 3 using the complete-case population of respondents.

Changes made: Results in column 2 & 3 of Table 3 updated [p13]

Comment 3: The number of missing values for ethnicity, long-standing conditions and time since first treatment are all not negligible at 7%, and it is not clear how removal of these observations might affect distributions for other variables. I would expect, therefore, some sensitivity multivariate analysis – either analysing missing values as separate categories or applying multiple imputations (whichever is more appropriate).

Response: We had previously conducted a sensitivity analysis but had not found it to be useful when interpreting the results and so had not included it in this manuscript. We accept the point that “it is not clear how removal of these observations might affect distributions for other variables.”

Changes made: Results added to [p11, line 42-53] and supplementary table 1 created detailing the distribution of variables in the case-complete population used for analysis [p25]

The distribution of patient, clinical and trust-level characteristics in the “complete-case” and “all respondents” populations was similar (Supplementary Table 1) and there was little difference in the univariate associations between the demographic characteristics and overall rating of care (with the exception of being deaf/having a hearing impairment which was not associated with a poorer rating of overall care during complete-case analysis, Table 3).

REVIEWER 2

Comment 1: The national patient experience survey suffers from all the limitations of national surveys of this sort but the high response rate has remained solid through repeated surveys. The study only looks at acute episodes of care and has a lower than prevalent response rate from classic "hard to reach" groups. It is not a new finding that there are inequalities in reported patient experience across gender, age and ethnicity but the fact that a national survey of this type is able to pick this up and demonstrate variation is of interest and I think worthy of publication.

Response: No response.

Changes made: None required.